

**Atmospheric Chemistry and Physics**

**Discussions**

# Using airborne HIAPER Pole-to-Pole Observations (HIPPO) to evaluate model and remote sensing estimates of atmospheric carbon dioxide

C. Frankenberg[1,2], S. S. Kulawik[3], S. Wofsy[4], F. Chevallier[5], B. Daube[4], E. A. Kort[6], C. O'Dell[7], E. T. Olsen[2], and G. Osterman[2]

[1]California Institute of Technology, Division of Geology and Planetary Sciences, Pasadena, CA, USA
[2]Jet Propulsion Laboratory, California Institute of Technology, Pasadena, CA, USA
[3]Bay Area Environmental Research Institute, Sonoma, CA 95476, USA
[4]Harvard University, Cambridge, MA, USA
[5]Laboratoire des Sciences du Climat et de l'Environnement (LSCE), Gif sur Yvette, France
[6]Climate and Space Sciences and Engineering, University of Michigan, Ann Arbor, Michigan, USA
[7]Cooperative Institute for Research in the Atmosphere (CIRA), Fort Collins, CO, USA

**ACPD**

doi:10.5194/acp-2015-961

**HIPPO model-satellite comparison**

C. Frankenberg et al.



Received: 25 November 2015 – Accepted: 9 December 2015 – Published: 19 January 2016

Correspondence to: C. Frankenberg (cfranken@caltech.edu)

Discussion Paper | Discussion Paper | Discussion Paper | Discussion Paper | Discussion Paper |

**ACPD**

doi:10.5194/acp-2015-961

**HIPPO model-satellite comparison**

C. Frankenberg et al.



**ACPD**

doi:10.5194/acp-2015-961

C. Frankenberg et al.



## Abstract

In recent years, space-borne observations of atmospheric carbon-dioxide ($CO_2$) have become increasingly used in global carbon-cycle studies. In order to obtain added value from space-borne measurements, they have to suffice stringent accuracy and precision requirements, with the latter being less crucial as it can be reduced by just enhanced sample size. Validation of $CO_2$ column averaged dry air mole fractions ($XCO_2$) heavily relies on measurements of the Total Carbon Column Observing Network TC-CON. Owing to the sparseness of the network and the requirements imposed on space-based measurements, independent additional validation is highly valuable. Here, we use observations from the HIAPER Pole-to-Pole Observations (HIPPO) flights from January 2009 through September 2011 to validate $CO_2$ measurements from satellites (GOSAT, TES, AIRS) and atmospheric inversion models (CarbonTracker CT2013B, MACC v13r1). We find that the atmospheric models capture the $XCO_2$ variability observed in HIPPO flights very well, with correlation coefficients ($r^2$) of 0.93 and 0.95 for CT2013B and MACC, respectively. Some larger discrepancies can be observed in profile comparisons at higher latitudes, esp. at 300 hPa during the peaks of either carbon uptake or release. These deviations can be up to 4 ppm and hint at misrepresentation of vertical transport.

Comparisons with the GOSAT satellite are of comparable quality, with an $r^2$ of 0.85, a mean bias $\mu$ of $-0.06$ ppm and a standard deviation $\sigma$ of 0.45 ppm. TES exhibits an $r^2$ of 0.75, $\mu$ of 0.34 ppm and $\sigma$ of 1.13 ppm. For AIRS, we find an $r^2$ of 0.37, $\mu$ of 1.11 ppm and $\sigma$ of 1.46 ppm, with latitude-dependent biases. For these comparisons at least 6, 20 and 50 atmospheric soundings have been averaged for GOSAT, TES and AIRS, respectively. Overall, we find that GOSAT soundings over the remote pacific ocean mostly meet the stringent accuracy requirements of about 0.5 ppm for space-based $CO_2$ observations.

# 1 Introduction

Space-borne measurements of atmospheric carbon dioxide can provide unique constraints on carbon exchanges between land, ocean, and atmosphere on a global scale. Results from the Scanning Imaging Absorption Spectrometer for Atmospheric CHartography SCIAMACHY (e.g. Schneising et al., 2014) and the Greenhouse Gas Observing Satellite GOSAT (Lindqvist et al., 2015) haven shown to reproduce the seasonal cycle as well as the secular trend of total column $CO_2$ abundances reasonably well (Kulawik et al., 2015). However, accuracy requirements are very stringent (Miller et al., 2007), warranting large scale biases of less than 0.5–1 ppm, being less than 0.3 % of the global background concentration. This is one of the most challenging remote sensing measurement from space as we not only want to reproduce known average seasonal cycles and trends but also small inter-annual deviations, resolved to subcontinental scales. There have been successes in doing so (e.g. Basu et al., 2014; Guerlet et al., 2013) but controversies regarding overall retrieval accuracy on the global scale still remain (Chevallier, 2015) and can neither be fully refuted nor confirmed with validations against the Total Column Carbon Observing Network (TCCON) (e.g. Kulawik et al., 2015). In addition, total uncertainties might be a mix of measurement and modeling biases (Houweling et al., 2015), for which uncertainties in vertical transport can play a crucial role (Stephens et al., 2007; Deng et al., 2015).

In this manuscript, we use the term accuracy to refer to systematic errors that remain after infinite averaging and can vary in space and time. Globally constant systematic errors are easy to correct for but those with spatio-temporal dependencies can have a potentially large impact on flux inversions.

Given the importance of the underlying scientific questions regarding the global carbon cycle and the challenging aspect of both the remote sensing aspect as well as the atmospheric inversion, every additional independent validation beyond ground-based data can be crucial. Here, we use measurements from the HIAPER Pole-to-Pole Ob-

Discussion Paper | Discussion Paper | Discussion Paper | Discussion Paper | Discussion Paper

**ACPD**

doi:10.5194/acp-2015-961

**HIPPO model-satellite comparison**

C. Frankenberg et al.

servations (HIPPO) programme (Wofsy, 2011) to evaluate both atmospheric models as well as remotely sensed estimates of atmospheric $CO_2$.

## 2   Data description

### 2.1   HIPPO

The HIAPER Pole-to-Pole Observations (HIPPO) project, a sequence of five global aircraft measurement programs, sampling the atmosphere from (almost) the North Pole to the coastal waters of Antarctica, from the surface to 14 km, spanning the seasons (Wofsy, 2011). This enables a comparison of both individual sub-columns of air but also integrating the atmosphere across the troposphere, which dominates variability in the column-averaged mixing ration of $CO_2$, denoted $XCO_2$. The campaigns covered different years as well as different seasons, namely: HIPPO 1: 8 January–30 January 2009, HIPPO 2: 31 October–22 November 2009, HIPPO 3: 24 March–16 April 2010, HIPPO 4: 14 June–11 July 2011, HIPPO 5: 9 August–9 September 2011.

Figure 1 shows an overview of the locations of the HIPPO profiles taken during different campaigns. As the 5 campaigns covered the years 2009 through 2011, we normalized the latitudinal cross section plot by subtracting the average $XCO_2$ around 50° south. In the Southern Hemisphere, the shape of the latitudinal gradients only changes marginally between seasons while the amplitude at the higher latitudes in the north spans about 10 ppm, with the strongest drawdown during August/September for HIPPO 5 and the highest concentrations during HIPPO 3 in March/April The dataset thus covers a wide range of atmospheric $CO_2$ profiles especially in the Northern Hemisphere where the strong biogenic cycle causes strong seasonality in $CO_2$ fluxes.

### 2.2   Atmospheric models

For the comparison of HIPPO against model data as well as for a more robust comparison of HIPPO against total column satellite $CO_2$ observations, we use two independent

**ACPD**

doi:10.5194/acp-2015-961

**HIPPO model-satellite comparison**

C. Frankenberg et al.

atmospheric models that both provide 4-D $CO_2$ fields (space and time) that are consistent with in-situ measurements of atmospheric $CO_2$. The main differences between those are the use of a different inversion scheme as well as underlying transport model. In addition, both models were used to extend individual HIPPO profiles from the highest flight altitude to the top of atmosphere when comparing to total column estimates from the satellite.

### 2.2.1 CarbonTracker CT2013B

CarbonTracker (Peters et al., 2007, with updates documented at http://carbontracker. noaa.gov) is a $CO_2$ modeling system developed by the NOAA Earth System Research Laboratory. CarbonTracker (CT) estimates surface emissions of carbon dioxide by assimilating in situ data from NOAA observational programs, monitoring stations operated by Environment Canada, and numerous other international partners using an ensemble Kalman filter optimization scheme built around the TM5 atmospheric transport model (Krol et al., 2005; http://www.phys.uu.nl/~tm5/). Here we use the latest release of CarbonTracker, CT2013B, which provides $CO_2$ mole fraction fields globally from 2000–2012. In this study, we interpolate modeled $CO_2$ mole fractions to the times and locations of individual HIPPO observations.

### 2.2.2 MACC v13r1

Monitoring Atmospheric Composition and Climate (MACC, http://www. copernicus-atmosphere.eu/) is the European Union-funded project responsible for the development of the pre-operational Copernicus atmosphere monitoring service. Its $CO_2$ atmospheric inversion product relies on a variational Bayesian formulation, developed by LSCE, that estimates 8 day grid-point daytime/nighttime $CO_2$ fluxes and the grid point total columns of $CO_2$ at the initial time step of the inversion window. It uses the global tracer transport model LMDZ (Hourdin et al., 2006), driven by the wind analyses from the ECMWF. Version 13r1 of the product covers the period from 1979 to

Discussion Paper | Discussion Paper | Discussion Paper | Discussion Paper |

**ACPD**

doi:10.5194/acp-2015-961

**HIPPO model-satellite comparison**

C. Frankenberg et al.

2013, at horizontal resolution 3.75° × 1.9° (longitude–latitude). It assimilated the dry air mole fraction measurements from 131 $CO_2$ stations over the globe in a unique 35 year assimilation window (see the list of sites in Tables S1 and S2 of Chevallier 2015). For this study, the model simulation has been interpolated to the time and location of the individual observations using the subgrid parametrization of the LMDZ advection scheme in the 3 dimensions of space (Hourdin and Armengaud, 1999). For the sake of brevity, we refer to MACC version 13r1 simply as MACC.

## 2.3 Satellite data

We use remotely sensed $CO_2$ observations from three different instruments, namely GOSAT, the Thermal Emission Sounder TES and the Atmospheric Infrared Sounder AIRS. As most HIPPO profiles took place over the oceans, SCIAMACHY was not included in the analysis. While GOSAT $CO_2$ is representative of the column averaged dry mole fraction ($XCO_2$), both TES and AIRS are most sensitive to the atmosphere around 500 and 300 hPa, respectively.

### 2.3.1 GOSAT (ACOS B3.5)

GOSAT takes measurements of reflected sunlight in three short-wave bands with circular footprints (diameter of 10.5 km) at nadir (Hamazaki et al., 2005; Kuze et al., 2009). Science data is starting in July 2009. In this work, we use column averaged dry air mole fraction ($XCO_2$) retrievals produced by NASA's Atmospheric $CO_2$ Observations from Space (ACOS) project, version 3.5 (see O'Dell et al., 2012 for retrieval details), which is very similar to the B3.4 version described in https://co2.jpl.nasa.gov/static/docs/v3.4_DataUsersGuide-RevB_131028.pdf. The data and bias correction as used here is identical to the dataset investigated in Kulawik et al. (2015).

Discussion Paper | Discussion Paper | Discussion Paper | Discussion Paper |

**ACPD**

doi:10.5194/acp-2015-961

**HIPPO model-satellite comparison**

C. Frankenberg et al.

**ACPD**

doi:10.5194/acp-2015-961

**HIPPO
model-satellite
comparison**

C. Frankenberg et al.

### 2.3.2 TES

TES is on the Earth Observing System Aura (EOS-Aura) satellite and makes high spectral resolution nadir measurements in the thermal infrared (660–2260 $cm^{-1}$, with un-apodized resolution of 0.06 $cm^{-1}$, apodized resolution of 0.1 $cm^{-1}$). TES was launched in July 2004 in a sun-synchronous orbit at an altitude of 705 km with an equatorial crossing time of 13:38 (local mean solar time) and with a repeat cycle of 16 days. From September 2004 through June 2011, TES collected "global survey" observations, averaging $\approx$ 500 good quality $CO_2$ day/night and land/ocean observations with cloud optical depth less than 0.5 between 40° S and 45° N. The peak sensitivity of $CO_2$ is about 500 hPa, with full-width half-maximum sensitivity between 200 and 800 hPa. TES $CO_2$ requires averaging to reduce random and systematic errors, with errors $\approx$ 6 ppm for a single observation to $\approx$ 1.3 ppm for monthly regional scales. For more details on TES $CO_2$, see Kulawik et al. (2013).

### 2.3.3 AIRS (v5)

The AIRS Version 5 (V5) tropospheric $CO_2$ product is a retrieval of the weighted partial-column dry volume mixing ratio characterizing the mid- to upper-tropospheric $CO_2$ concentration. The product is derived by the technique of Vanishing Partial Derivatives (VPD) described in Chahine et al. (2005) and is reported at a nominal nadir resolution of 90 km × 90 km over the globe over the latitude range 60° S to 90° N and time span September 2002 to present.

The VPD method assumes a $CO_2$ profile that is a linearly time-dependent global average constant volume mixing ratio throughout the atmosphere. Using that prior profile, the VPD derives $CO_2$ by shifting the $CO_2$, $T$, $q$ and $O_3$ profiles and minimizing the residuals between the cloud-cleared radiances and those resulting from the forward calculation for channel subsets selected to avoid contamination by surface emission (except in regions of high topography). Further, it localizes the maximum sensitivity to variations of $CO_2$ concentration to the pressure regime spanning 300 to 700 hPa.

In normal practice, the AIRS Level 2 products ingested by the $CO_2$ post-processing retrieval stage are retrieved using the combination of the infrared instrument and a companion Advanced Microwave Sounding Unit (AMSU). The 5–7 year expected lifetime of AMSU based on NOAA experience is much shorter than that of the AIRS instrument, so an alternate Level 2 retrieval using only the infrared radiances (AIRS_Only) was developed. The VPD retrieval normally ingests the combined IR/MW retrieval system products. Beginning in January 2011 the degradation of AMSU channel 5 noise figure significantly reduced the IR/MW L2 product yield so that the ingest was shifted to the IR-Only L2 product. Validation against HIPPO measurements of the $CO_2$ retrievals resulting from ingesting IR/MW L2 and IR-Only L2 products indicates that the products are equivalent (Olsen and Licata, 2014).

## 3   HIPPO – Model inter-comparisons

Figure 2 shows an overview of model-HIPPO differences at 3 pressure levels as well as $XCO_2$, the total column average. For the differences in $XCO_2$, the respective model has been used to extend the HIPPO profiles from its highest altitude to the top of atmosphere, hence part of the smaller differences observed in $XCO_2$ comparisons can stem from the fact that the model contributes slightly to the HIPPO based $XCO_2$ as well, though the tropospheric variability should dominate. As can be seen in the left panels, not all HIPPO profiles extend up to 300 hPa.

Unsurprisingly, model-data mismatches at individual levels are substantially higher than in the total column, about a factor 2. Many differences are not consistent between model, for example during HIPPO 4N, extending from West Papua northwards. In MACC, there is first a substantial underestimation throughout the profile and then an overestimation further north. In CT2013B, no obvious discrepancies can be observed. In other areas, such as the same HIPPO 4N path south of Alaska, MACC appears rather consistent but CT2013B is much higher at 800 hPa but much lower at 500 hPa, with a slight underestimate in the total column.

**ACPD**

doi:10.5194/acp-2015-961

**HIPPO model-satellite comparison**

C. Frankenberg et al.

Figure 3 provides an in-depth review of HIPPO – model comparisons for profiles averaged by latitudinal bands and campaign. In most cases, profiles agree to within 1 ppm with a few notable exceptions, mostly at higher latitudes during the drawdown or respiration maximum in HIPPO 5 and 3, respectively. These are typically associated with steep vertical gradients around 300 hPa, both in HIPPO 5 and 3, albeit with different signs. In most other cases, the differences even in the profiles are usually below 1 ppm, underlining the stringent accuracy requirements for space based $CO_2$ measurements, as atmospheric models optimized with respect to the ground-based network already model oceanic background concentrations fairly well. However, the caveat is that also these ground-based stations are located in remote regions, ideally not affected by local sources. On smaller spatial scales near sources, space-based measurements can provide valuable information even in the presence of potential large-scale biases.

Figure 4 shows an in-depth comparison of the largest model-HIPPO discrepancies, namely the high latitude profiles during HIPPO 3 and 5. As one can see on the left panels, the seasonal cycles in the mid-troposphere and at 200 hPa can be opposite, with large $CO_2$ values in the upper atmosphere during the largest $CO_2$ drawdown and vice versa during the peak of respiration. Model-HIPPO mismatches are most obvious and similar between models in HIPPO 3 (March/April 2010), with differences reaching up to 4 ppm at 300 hPa. This is consistent with a comparison against the GEOS-Chem model by Deng et al. (2015), who studied the impact of discrepancies in stratosphere–troposphere exchange on inferred sources and sinks of $CO_2$. It can be seen that most models suffer from these potential biases.

Overall, both CT2013B as well as MACC show an excellent agreement with HIPPO over the oceans. In some cases, MACC seems to compare somewhat better, which might be related to the longer inversion window of MACC, which can have an impact over remote areas such as the Pacific Ocean. However, this statement cannot be generalized as it may be specific to remote areas with low measurement density and be very different elsewhere.

**ACPD**

doi:10.5194/acp-2015-961

**HIPPO model-satellite comparison**

C. Frankenberg et al.



Discussion Paper | Discussion Paper | Discussion Paper | Discussion Paper | Discussion Paper |

**ACPD**

doi:10.5194/acp-2015-961

**HIPPO model-satellite comparison**

C. Frankenberg et al.

## 4 Comparisons of column-averaged mixing ratios

Here, we look at column-averaged dry air mole fractions $XCO_2$, derived using absorption spectroscopy of reflected sun-light recorded by near-infrared spectrometers such as SCIAMACHY, GOSAT or OCO-2. In this paper, we only used GOSAT data as it is the only instrument having sampled in Glint mode during the HIPPO investigation. SCIAMACHY data over the oceans is not yet matured as is has no dedicated Glint mode. OCO-2 could largely improve on GOSAT's data density over the oceans but didn't overlap with the HIPPO measurement campaign period. The new Atmospheric Tomography Mission (ATom), selected as one of NASA's Earth Venture airborne missions, will potentially allow for similar comparisons to OCO-2 in the future.

For the comparison of column-averaged mixing ratios, we need to extend the HIPPO profiles to the top-of-atmosphere. For this, we use the respective atmospheric model to compare with. In addition, we computed the average $XCO_2$ for each campaign using all the data and subsequently removed it from individual measurements. This ensures that observed correlations are driven pre-dominantly by spatial gradients within a campaign period and not by the secular trend. For satellite data, we include the instrument sensitivity by applying the averaging kernel to the measured profile (in other words, this conversion computes what the respective instruments *should* measure if HIPPO were the truth).

### 4.1 Atmospheric models

In terms of $XCO_2$, both atmospheric models used here compare extremely well against HIPPO. Even after normalization with the campaign average, the correlation coefficients and slopes are $r^2 = 0.93$ (slope = 0.95) for CT2013B and $r^2 = 0.95$ (slope = 1.00) for MACC. South of 20° N, almost all data-points lie within ±1 ppb with some outliers of up to 3 ppb at higher latitudes, mostly over the continents (see Fig. 2).

These numbers should not be used to compare the models against each other because, as evident in Fig. 2, there are regions where either one or the other model is

in better agreement with HIPPO. In conclusion, one can state that most model mismatches are below 1 ppm in remote areas such as the oceans and can reach 2–3 ppm over the continents with potentially higher values in under-sampled areas with high $CO_2$ uptake such as the US corn belt. In addition, it should be mentioned that both models ingest a multitude of $CO_2$ measurements at US ground-based stations and areas further away might be less well modeled. However, the excellent agreement provides a benchmark against which satellite retrievals have to be measured.

## 4.2  GOSAT

The comparison of GOSAT satellite data against HIPPO is somewhat more complicated because there is not necessarily a matching GOSAT measurement with each HIPPO profile. For coincidence criteria, we follow exactly Kulawik et al. (2015), based on the dynamic co-location criteria detailed in Wunch et al. (2011); Keppel-Aleks et al. (2011, 2012). In addition, we require that the difference of CT2013B sampled at the HIPPO and the actual GOSAT location is less than 0.5 ppm, thereby bounding the error introduced by the spatial mismatch between HIPPO and respective GOSAT soundings. For each match, the standard error in the GOSAT $XCO_2$ average is computed using the standard deviation of all corresponding GOSAT colocations divided by the square root of the number of colocations.

For the GOSAT comparison, we require more than 5 co-located GOSAT measurement per HIPPO profile. HIPPO $XCO_2$ is computed as the average of MACC and CT2013B extended HIPPO profiles with the difference between the two used as uncertainty range for HIPPO.

In Fig. 7, the scatterplot of HIPPO vs. GOSAT is depicted. It is obvious that the data density is far lower than for the models because (a) HIPPO 1 is not overlapping in time and (b) only a subset of HIPPO profiles is matched with enough co-located GOSAT soundings. This gives rise to a reduced dynamic range in $XCO_2$, from about −1.5 to 3 ppm difference to the campaign average. However, both slope and $r^2$ are also in excellent agreement with HIPPO and only very few points are exceeding 1 ppm differ-

**[ACPD](doi:10.5194/acp-2015-961)**

doi:10.5194/acp-2015-961

**HIPPO model-satellite comparison**

C. Frankenberg et al.

ence. Those that are $< -1$ ppm are also associated with larger uncertainties induced by model extrapolation, as seen in the larger error-bars for HIPPO in the left panel (esp. for HIPPO 2S). The right panel shows the discrepancies for the models as well, just for the subset that could be compared against GOSAT and using the model sampled at

the GOSAT locations.

One can see that it is hard to make a clear statement on whether GOSAT or the models compare better with HIPPO. Figure 8 shows this comparison in more detail, plotting model-HIPPO differences on the $x$ axis and GOSAT-model differences on the $y$ axis. As before, the error-bar for GOSAT is derived as the standard error in the mean

and the model error-bar by using the variability of HIPPO XCO2 using the 2 different models to extrapolate to the top-of-atmosphere (and the average of the 2 is defined as HIPPO XCO$_2$. The center box spans the range from –0.5–0.5 ppm, a strict requirement for systematic biases (GHG-CCI, 2014). The green and red shaded areas indicated regions where either the GOSAT data meets the 0.5 ppm requirement but the models

not (green) or vice versa (red). Given the small amount of samples, it is premature to draw strong conclusions but it appears that somewhat more points lie in the green area. It also has to be pointed out that pure measurement unsystematic noise also contributes to the scatter in GOSAT.

For MACC, there is even a noticeable correlation between MACC-HIPPO and

GOSAT-HIPPO with an $r^2$ of 0.26. This can hint at either small-scale features caught by HIPPO and missed by both GOSAT and models or small systematic variability between the exact HIPPO and GOSAT co-location. Most of the samples causing the high $r^2$ are located in the lower left quadrant, underestimated by GOSAT and both models and apparently all within HIPPO 2S, located between 40S and 20S.

Figure 9 depicts the HIPPO 2S campaign in more detail, showing the exact flight patterns and the differences with respect to MACC (MACC-HIPPO) at each measurement point (upper panel). For the sake of simplicity, we only show MACC here. The measured CO concentrations are shown in the lower panel. There is enhanced Carbon Monoxide (CO) at higher altitudes, indicating long-range transport of biomass burn-

**ACPD**

doi:10.5194/acp-2015-961

**HIPPO model-satellite comparison**

C. Frankenberg et al.

ing at the time of overflight, which can explain the apparent model-HIPPO mismatch. The features span several degrees of latitude, excluding coarse model resolution as a reason for missing the plume. Thus, we hypothesize that the mismatch is caused by either understimated CO emissions from the GFED (Randerson et al., 2013) emission database (which is used by both models) or transport errors in the models. For GOSAT, the mismatch is most likely caused by too lenient coincidence criteria, missing most of the biomass burning plume.

Overall, it can be concluded that GOSAT measurements can provide valuable and accurate information on the global $CO_2$ distribution and meets the 0.5 ppm bias criterion in most cases over the ocean. However, small sampling sizes precludes an in-depth analysis of potential large-scale biases in the datasets. In the future, OCO-2 with its much higher sampling density will help to disentangle measurement and modeling bias and guide inversion studies.

## 5  Comparisons of mid to upper tropospheric $CO_2$

### 5.1  TES (∼ 510 hPa)

For the comparison with TES, we use the 510 hPa retrieval layer and apply averaging kernels accordingly. Coincidence criteria are identical to the GOSAT analysis but we require at least 20 valid TES soundings per HIPPO profile to reduce measurement noise. Similar to before, the TES error-bars are empirically derived using the standard deviation of the co-located soundings itself.

Figure 10 shows the comparison of TES against HIPPO in the same way as done for GOSAT. The correlation ($r^2$) is somewhat lower than for GOSAT but still very significant. Some differences exceed 2 ppm, albeit with a relatively high standard error, i.e. barely significant at the $2\sigma$ level (see right panel, error-bars indicate $1\sigma$).

Given the larger standard error in TES data, differences may be purely noise driven and not necessarily a hint at large-scale biases even though the clustering of positive

**[ACPD](ACPD)**

doi:10.5194/acp-2015-961

**HIPPO model-satellite comparison**

C. Frankenberg et al.

Discussion Paper | Discussion Paper | Discussion Paper | Discussion Paper | Discussion Paper

anomalies, esp. in HIPPO 3 at higher latitudes, is apparent. As evident from Fig. 3, there are stronger vertical gradients at 15–45° N during HIPPO3 because they are close to the peak $CO_2$ value caused by wintertime respiration. This can cause potential mismatches as gradients can be strong and co-location criteria might have to be more
strict. In addition, the HIPPO profiles are extended by models to the top-of-atmosphere and are thus not entirely model-independent.

## 5.2  AIRS (~ 300 hPa)

For the comparison with AIRS, the sensitivity maximum varies around 300 hP and we apply the averaging kernels accordingly. Owing to the large data density and high sin-
gle measurement noise of AIRS, we use a minimum of 50 colocations for a comparison, still leaving many more data-points than for the GOSAT and TES comparison. As coincidence criteria, we use data within 5° latitude and longitude and 24 h time difference.
     Even though the correlations are significant, a bias dependence on latitude can be observed, which hampers incorporation of AIRS data into flux inversions. The reason
for these biases is currently unknown but may be related to changes in peak sensitivity altitude as a function of latitude. A full characterization of averaging kernels per sounding would alleviate these concerns. Given the observed larger model-HIPPO $CO_2$ differences at higher altitudes, a fully characterized AIRS $CO_2$ product could be worthwile for the flux community. However, requirements for systematic biases in partial columns
are even stricter than for the total column (Chevallier, 2015).

## 6  Conclusions

In this study, we compared atmospheric models as well as satellite data of $CO_2$ against HIPPO profiles. Table 1 provides a high level overview of the derived statistics. Both atmospheric models compare very similarly, both showing a very high correlation with
respect to HIPPO, even with subtracting the campaign average $XCO_2$, as is done

Discussion Paper | Discussion Paper | Discussion Paper | Discussion Paper |

**ACPD**

doi:10.5194/acp-2015-961

**HIPPO model-satellite comparison**

C. Frankenberg et al.

throughout all comparisons. Largest discrepancies are found near 300 hPa at higher latitudes during peak wintertime $CO_2$ accumulation as well as the summer uptake period. These may be related to steep vertical gradients poorly resolved by the models. In addition, a biomass burning event in the Southern Hemisphere seems to have been underestimated by the models, causing discrepancies of around 1 ppm.

For GOSAT comparisons, results are comparable but the sample size is much smaller. A comparison of GOSAT and model mismatches with respect to HIPPO indicates that GOSAT compares slightly better overall. In the future, OCO-2 with its much higher sampling density and expanded latitudinal coverage over the oceans should provide enough data to draw more robust conclusions that using GOSAT, for which the data density is fairly low. In general, GOSAT compares very well to HIPPO, followed by TES and AIRS. For TES, most deviations can be explained by pure measurement noise but AIRS appears to exhibit some latitudinal biases that would need to be accounted for if used for source-inversion studies. On the other hand, systematic model transport errors that can affect source inversions (Deng et al., 2015) were confirmed here for both atmospheric models used. Despite initial scepticism towards using remotely sensed $CO_2$ data for global carbon cycle inversion, we are now reaching a state where potential systematic errors in both remote sensing as well as atmsopheric modeling can play en equally crucial part. Innovative methods to characterize and ideally minimize both of these error sources will be needed in the future. One option is to apply flux inversion schemes that co-retrieve systematic biases alongside fluxes, such as in Bergamaschi et al. (2007), using prior knowledge on potential physical insight into systematic biases, such as aerosol interference, land/ocean biases or air mass factors.

*Acknowledgements.* Funded by NASA Roses ESDR-ERR 10/ 10-ESDRERR10-0031, "Estimation of biases and errors of $CO_2$ satellite observations from AIRS, GOSAT, SCIAMACHY, TES, and OCO-2". We thank the entire HIPPO team for making these measurements possible and the NIES and JAXA GOSAT teams for designing and operating the GOSAT mission and generously sharing L1 data with the ACOS project.

**ACPD**

doi:10.5194/acp-2015-961

**HIPPO model-satellite comparison**

C. Frankenberg et al.

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

**Table 1.** Summary of all HIPPO comparisons. $\#_{profiles}$ shows how many HIPPO profiles were used for the comparison. Correlation coefficients, fitted slope, mean difference $\mu$ and standard deviation $\sigma$ of different compared to HIPPO of all comparisons are computed using measurements normalized by the respective campaign average. For comparison, $\sigma$ of model-HIPPO for the satellite colocations and respective sensitivity are provided as well.

|  | $\#_{profiles}$ | $r^2$ | slope | $\mu$ (ppb) | $\sigma$ (ppb) | $\sigma_{CT}$ | $\sigma_{MACC}$ |
|---|---|---|---|---|---|---|---|
| GOSAT | 94 | 0.85 | 0.99 | −0.06 | 0.45 | 0.42 | 0.36 |
| TES | 135 | 0.75 | 1.45 | 0.34 | 1.13 | 0.36 | 0.3 |
| AIRS | 200 | 0.37 | 0.66 | 1.11 | 1.46 | 0.63 | 0.47 |
| CT2013B | 676 | 0.93 | 0.95 | 0.10 | 0.51 | n/a | n/a |
| MACC | 674 | 0.95 | 1.00 | 0.06 | 0.43 | n/a | n/a |

Discussion Paper | Discussion Paper | Discussion Paper | Discussion Paper |

**ACPD**

doi:10.5194/acp-2015-961

**HIPPO model-satellite comparison**

C. Frankenberg et al.

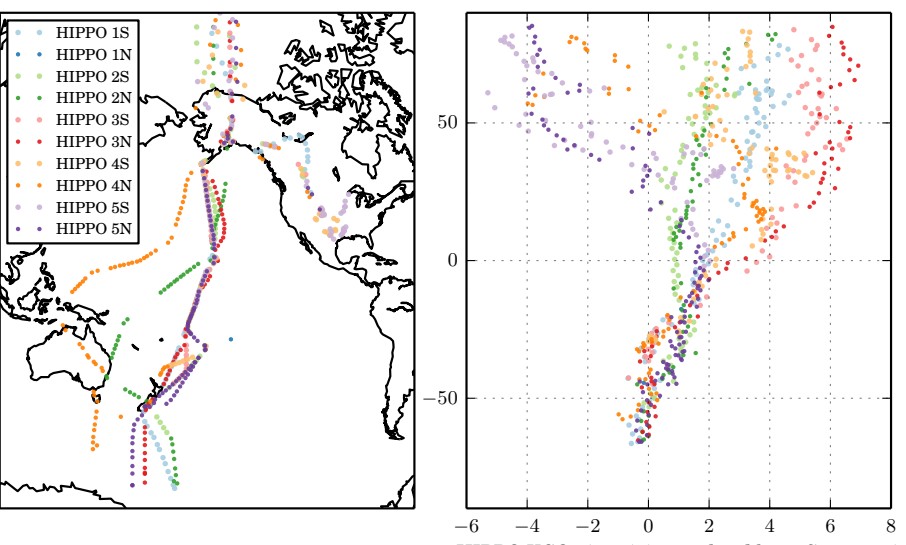

**Figure 1.** Left: overview of the 5 HIPPO campaigns, taken place in January 2009 (1), November 2009 (2), March/April 2010 (3), June/July 2011 (4) and August/September 2011 (5). Campaigns are separated by Southbound (S) and Northbound (N) and each dot indicates a separate HIPPO vertical profile. Right: latitudinal gradients of column averaged $CO_2$ mixing ratios with the campaign average at 50S subtracted. Above the highest HIPPO flight altitude, profiles have been extended with CarbonTracker CT2013B in order to compute the column average.

**ACPD**

doi:10.5194/acp-2015-961

**HIPPO model-satellite comparison**

C. Frankenberg et al.

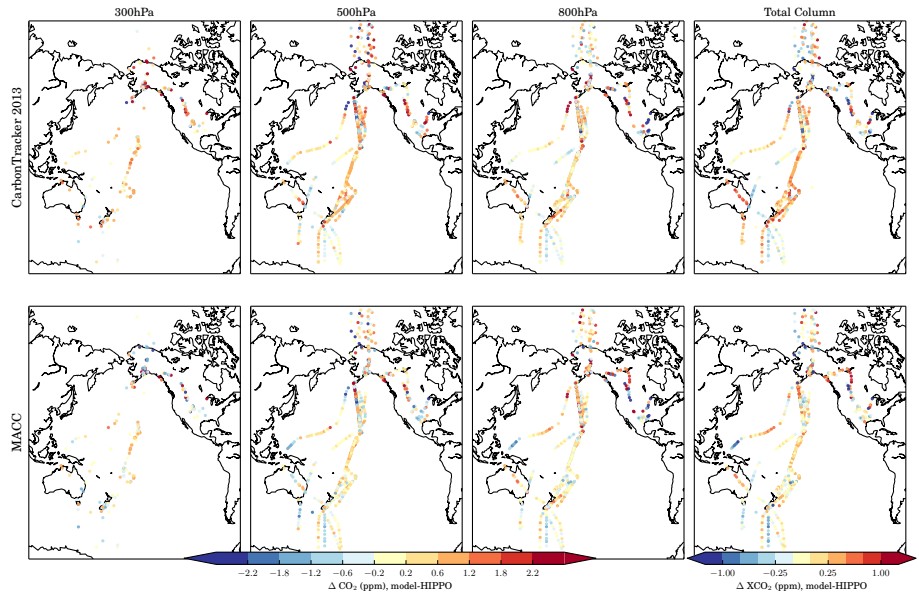

**Figure 2.** Top row, from left to right: CT2013B-HIPPO differences at 300, 500, 800 hPa and column averaged mixing ratio of $CO_2$. Bottom row: as top row but for the MACC model. Note the change in color-scale between layer and total column differences. All HIPPO campaigns are included.

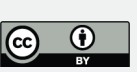

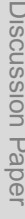

**ACPD**

doi:10.5194/acp-2015-961

**HIPPO model-satellite comparison**

C. Frankenberg et al.

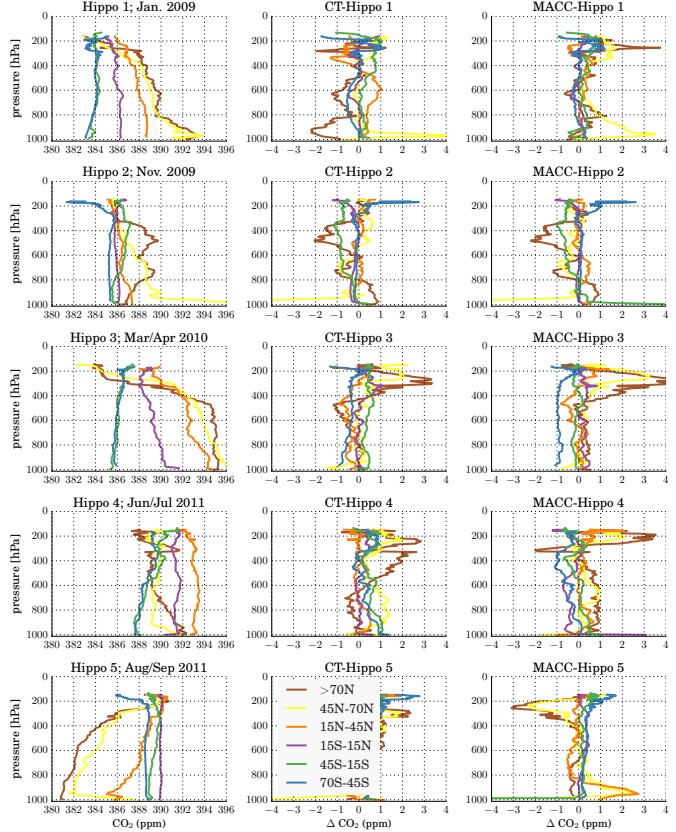

**Figure 3.** Summary of averaged $CO_2$ HIPPO profiles in ppm (left column) and model-HIPPO differences (middle and right column), separated by latitudinal bands (color-coded) and HIPPO campaign (separate rows).

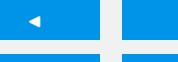

**ACPD**

doi:10.5194/acp-2015-961

**HIPPO model-satellite comparison**

C. Frankenberg et al.

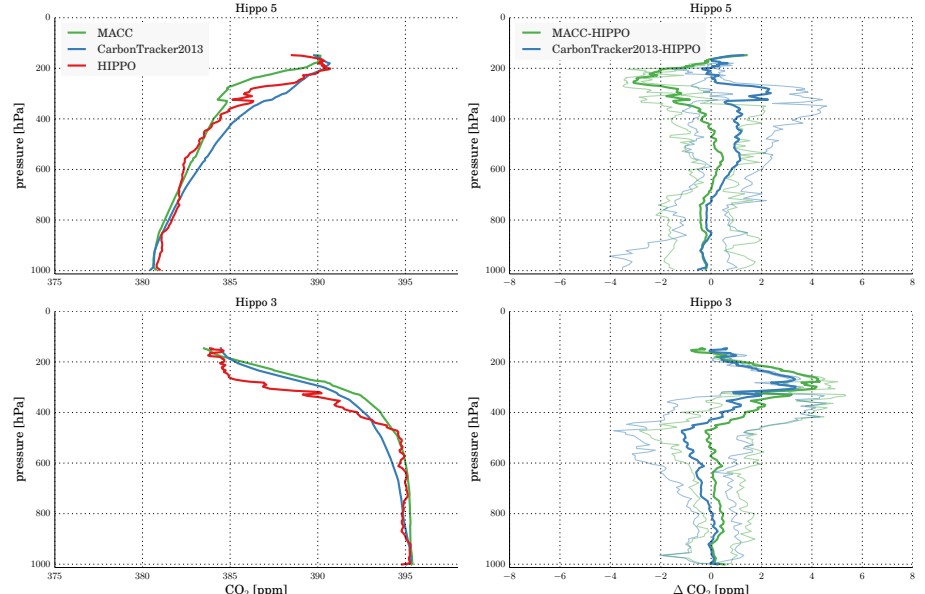

**Figure 4.** Averaged HIPPO and matched model profiles for latitudes > 70° N during HIPPO 3 and 5, respectively. The left panels shows model and HIPPO profiles and the right panels show model-HIPPO average differences as well as their range in the thinner and somewhat transparent colors.

**ACPD**

doi:10.5194/acp-2015-961

**HIPPO model-satellite comparison**

C. Frankenberg et al.

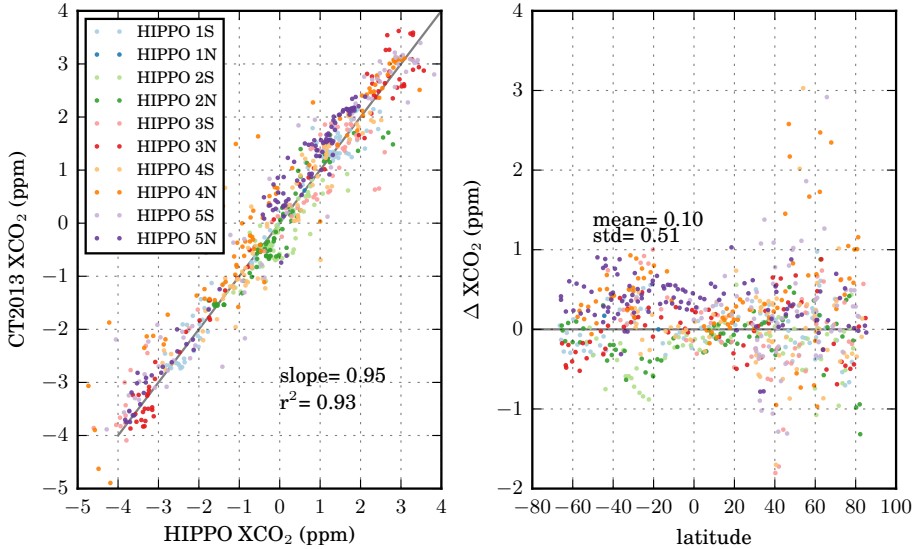

**Figure 5.** Left: scatterplot of XCO$_2$ computed from individual HIPPO profiles (*x* axis) against corresponding CT2013B data. Right: difference plot of XCO$_2$ against latitude. Campaigns as well as North and Southbound tracks are color-coded.

**ACPD**

doi:10.5194/acp-2015-961

**HIPPO model-satellite comparison**

C. Frankenberg et al.

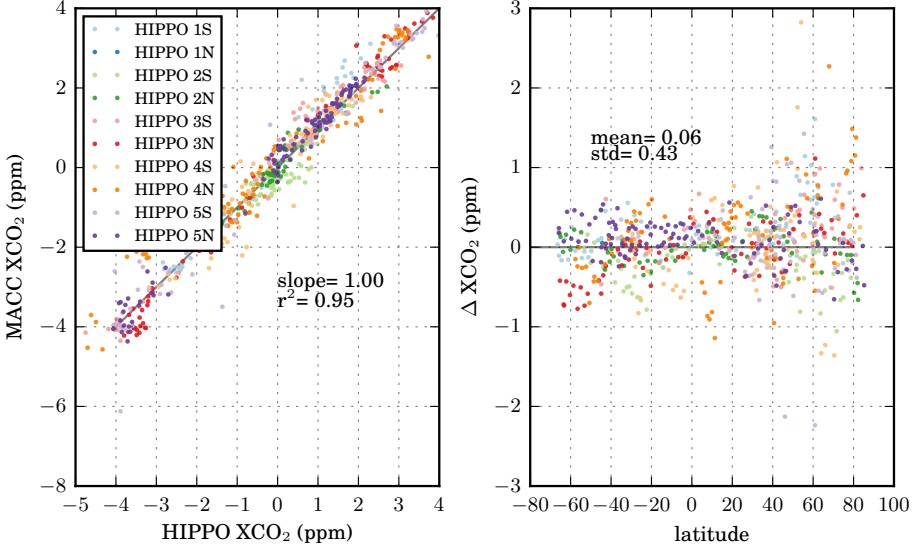

**Figure 6.** Left: scatterplot of $XCO_2$ computed from individual HIPPO profiles ($x$ axis) against corresponding MACC data. Right: difference plot of $XCO_2$ against latitude. Campaigns as well as North and Southbound tracks are color-coded.

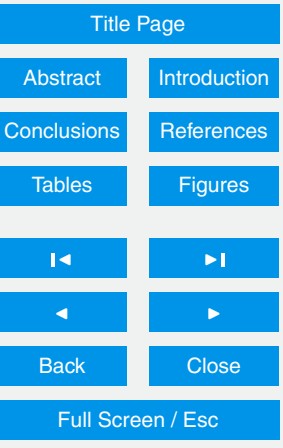

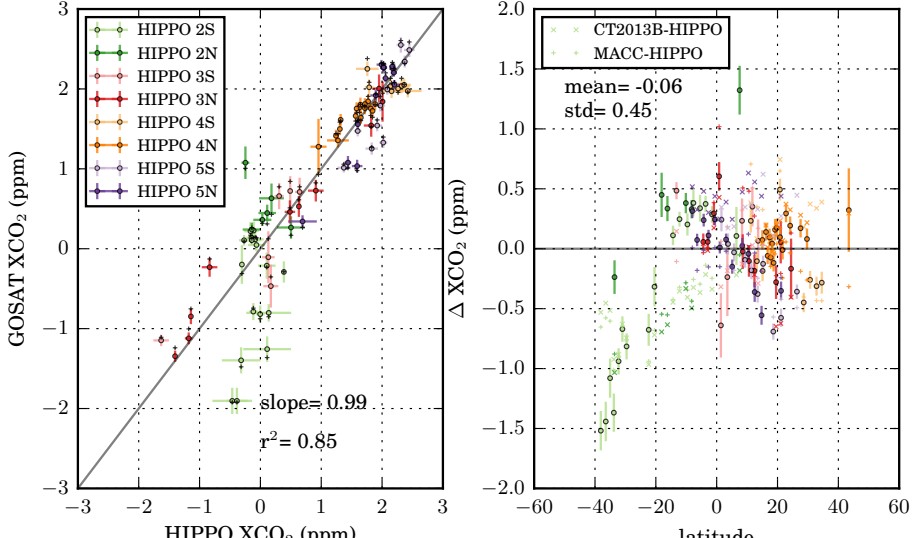

**Figure 7.** Left: scatterplot of $XCO_2$ computed from individual HIPPO profiles ($x$ axis) against corresponding GOSAT data. Right: difference plot of $XCO_2$ against latitude. Campaigns as well as North and Southbound tracks are color-coded. For comparison, the right panel also shows the model-HIPPO differences in smaller symbols without errorbar (MACC as +, CT2013B as x).

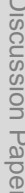

ACPD

doi:10.5194/acp-2015-961

**HIPPO model-satellite comparison**

C. Frankenberg et al.

**ACPD**

doi:10.5194/acp-2015-961

**HIPPO model-satellite comparison**

C. Frankenberg et al.

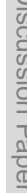

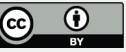

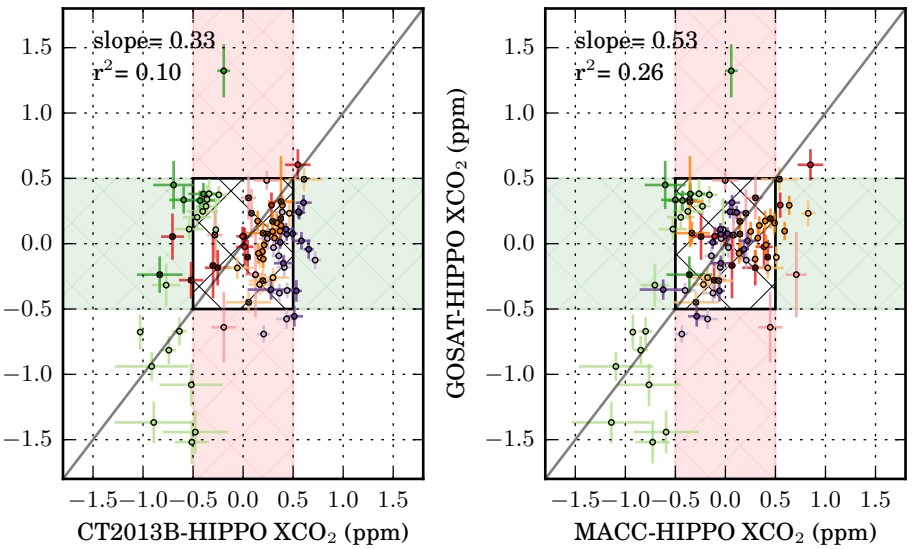

**Figure 8.** Left: scatterplot of $\Delta XCO_2$ (CT-HIPPO) against $\Delta XCO_2$ (GOSAT-HIPPO), using just the GOSAT subsets. Right: same as left but using MACC instead of CT2013B. The inner box represent the area where both model and GOSAT are within 0.5 ppm compared to HIPPO, which corresponds to the very stringent accuracy requirement. The green and red shaded areas correspond to regions where the satellite deviates less than the models and is within 0.5 ppm (green) as well as where the models deviate less than GOSAT (red). The white cells on the outer edges indicate areas where both deviate more than 0.5 ppm overall.

Discussion Paper | Discussion Paper | Discussion Paper | Discussion Paper |

**ACPD**

doi:10.5194/acp-2015-961

**HIPPO
model-satellite
comparison**

C. Frankenberg et al.



**Figure 9.** Top: MACC-HIPPO $CO_2$ differences (ppm) as a function of latitude and pressure level during the HIPPO 2 Southbound campaign, recorded on 10–11 November 2009. Bottom: corresponding HIPPO CO measurements (ppb).

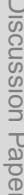

**ACPD**

doi:10.5194/acp-2015-961

**HIPPO model-satellite comparison**

C. Frankenberg et al.

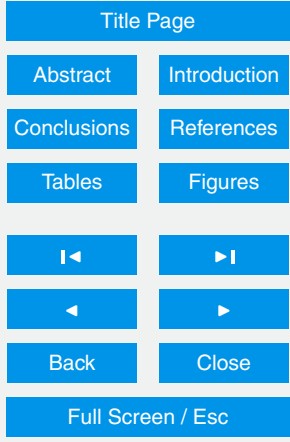

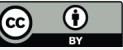

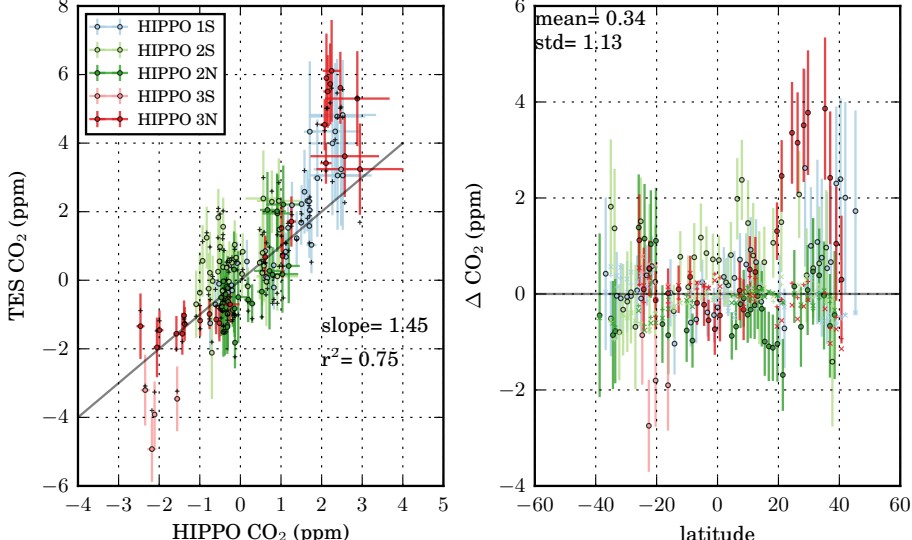

**Figure 10.** Left: scatterplot of $CO_2$ from individual HIPPO profiles ($x$ axis) against corresponding TES data. Right: difference plot of $CO_2$ against latitude. Campaigns as well as North and Southbound tracks are color-coded, model-HIPPO differences are plotted as well. Please refer to Fig. 7 for a detailed legend.

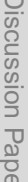

**ACPD**

doi:10.5194/acp-2015-961

**HIPPO
model-satellite
comparison**

C. Frankenberg et al.

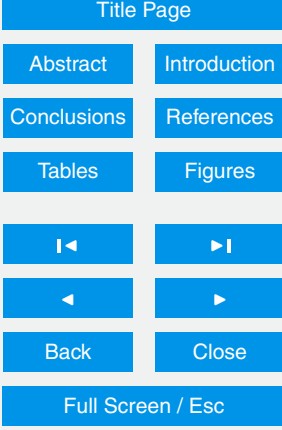

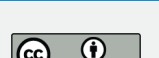

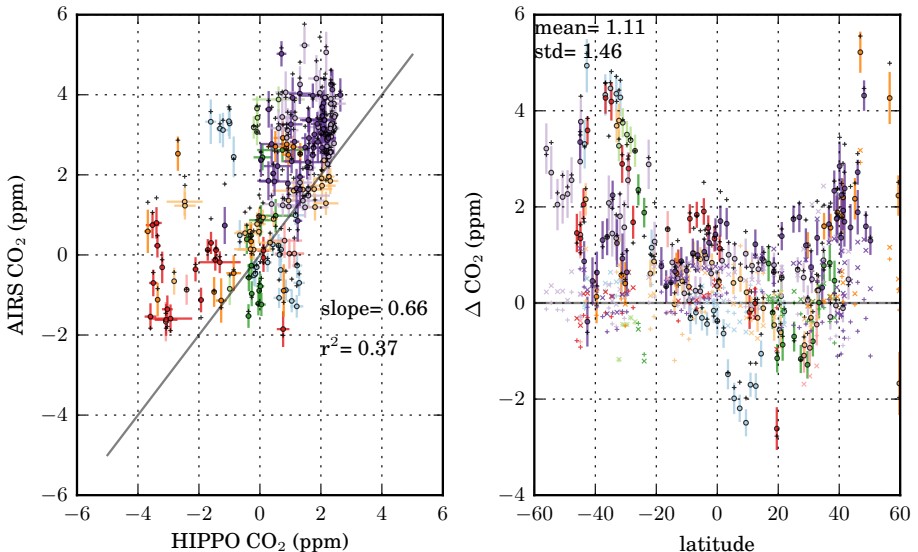

**Figure 11.** Left: scatterplot of $CO_2$ from individual HIPPO profiles (*x* axis) against correspond-ing AIRS data. Right: difference plot of $CO_2$ against latitude. Campaigns as well as North and Southbound tracks are color-coded, model-HIPPO differences are plotted as well. Please refer to Fig. 7 for a detailed legend.