# Peer review of "Using airborne HIAPER Pole-to-Pole Observations (HIPPO) to evaluate model and remote sensing estimates of atmospheric carbon dioxide"

_Atmospheric Chemistry and Physics, 2015_

## Referee Comment (RC1) · Anonymous Referee #1 · 10 Feb 2016

Manuscript "Using airborne HIAPER Pole-to-Pole Observations (HIPPO) to evaluate model and remote sensing estimates of atmospheric carbon dioxide" of Frankenberg et al. presents detailed comparisons of HIPPO aircraft $CO_2$ observations with different satellite-derived atmospheric $CO_2$ products (primarily GOSAT XCO2 but also AIRS and TES CO2) and two global models (MACC and CarbonTracker) . They show new interesting results, the topic is appropriate for ACP and the paper is well written. I recommend publication after the minor issues listed below have been considered by the authors (I have not identified any major issues).
[Figure]

Minor issues:

Page 10, line 24: I think "an excellent agreement" is a bit too strong taking into account that there are differences up to 4 ppm (same page, line 20). I recommend to change this to "good agreement" or so. Similar on page 11, line 21, with the statement "compare extremely well" already in the first sentence before any comparison results are shown and discussed. I also do not think that agreement within 1 ppm and outliers up to 3 ppm is best characterized by "extremely well" (NOTE: in the text on page 11 the unit ppb is given two times (lines 24 and 25) but I guess this should be ppm!).

Page 10, line 25: "In some cases, MACC seems to compare somewhat better, …". A MACC colleague is co-author but no CarbonTracker colleague. I wonder if NOAA would agree with this statement. I also wonder if NOAA needs to be acknowledged for their data.

Page 11, line 6: "SCIAMACHY data over the oceans is not yet matured as is has no dedicated Glint mode." Sounds a bit strange (even if "is" typo corrected). I recommend to replace this with "SCIAMACHY data have not been used as it has no dedicated glint mode and the SCIAMACHY products (e.g., Reuter et al., 2011) are limited to retrievals over land". Reuter et al., 2011: "Retrieval of atmospheric CO2 with enhanced accuracy and precision from SCIAMACHY: Validation with FTS measurements and comparison with model results", J. Geophys. Res.

Page 16, line 8: "… indicates that GOSAT compares slightly better overall." Compared to what?

Page 21, Tab. 1: Why is the GOSAT sigma only 0.45 ppm (as far as I know the GOSAT XCO2 single measurement precision is about 2 ppm; or have data been averaged?)? Please check and add additional explanation if necessary.

Fig. 3: Bottom, middle: Profiles only partially visible as overplotted by legend. Please improve.
[Figure]

Fig. 5, left: Possibly data points only partially visible as overplotted by legend. Please improve.

Fig. 6, left: Data points only partially visible as overplotted by legend. Please improve.

Fig. 7, right: Symbols for models very difficult to see in printout.

No reference to Figs. 5 and 6 in text (should be somewhere in Sect. 4).

No reference to Fig. 11 in text (should be somewhere in Sect. 5.2).

Typos:

Page 10, line 6: "are usually 162253" ?

Caption Tab. 1: "of different compared to" -> "of the difference compared to"
* * *

---

## Referee Comment (RC2) · Anonymous Referee #2 · 9 Mar 2016

The paper "Using airborne HIAPER Pole-to-Pole Observations (HIPPO) to evaluate model and remote sensing estimates of atmospheric carbon dioxide" by Frankenberg et al. describes the comparison of HIPPO aircraft $CO_2$ data with the CarbonTracker and MACC models and with the satellite instruments GOSAT, TES, and AIRS. Both total column dry air mole fractions and partial columns are considered. The emphasise of the study is to investigate how well models and satellite instruments perform in capturing background levels of methane in remote areas far away from local sources. The HIPPO campaigns are well-suited for these comparisons as the flights are largely over the oceans.

[Figure]

The paper is concisely written and provides new results and insights in the performances of various models and instruments. The topic and presentation is well suited for ACP and I would recommend publication.

I have only identified some minor issues, but there is one point the authors should address somewhat more extensive; the mathematical treatment of the comparison, ie. the application of the averaging kernels and the extensions of the HIPPO profiles above flight altitude (see below for details).

Minor revisions:

p5, l8-10: This enables ... denoted XCO2 I think this statement would be more clear to the reader when a line is added to indicate that an extension is needed above 14 km. Then you can indeed state that this extension is of limited consequence since most of the variability in XCO2 stems from the troposphere which is covered by the HIPPO profiles.

p7, l11-12: As most ... analysis. Add a line why SCIAMACHY does not provide data over oceans

p7, l16: short-wave –> short-wave infrared

p8, l11: How can averaging lead to the reduction of systematic errors?

p9, l9-11: Validation ... (Olsen and Licata, 2014). If Olsen and Licata already have compared IR/MW L2 and IR-Only L2 against HIPPO, then I would expect a sentence explaining how the current study differs and/or extends wrt. the cited paper.

p9, l14-18: For the differences ... should dominate If you first extend the HIPPO profile with model data, then integrate, and finally subtract the integrated model data, does the part above flight altitude not exactly cancel?

HIPPO (< 14 km) + model (> 14 km) - model (0-TOA) =

HIPPO (< 14 km) + model (> 14 km) - model (< 14 km) - model (> 14 km) =

HIPPO (< 14 km) - model (< 14 km).

So, I do not see how the extension can contribute to the difference between HIPPO and model.

p10, l14-23: Figure 4 ... potential biases. HIPPO 3 is nicely explained in this paragraph, but HIPPO 5 is depicted in the Figure but not mentioned. Any comment that the authors can make on the MACC and CT differences/similarities?

p11, l2-10: Here, we look ... in the future. This alinea is mostly about measurements and campaigns that are not treated in the paper. I understand why the authors like to mention this, but maybe the conclusion, which includes a future outlook mentioning OCO-2, is the better spot for this.

p11, l11-19: For the comparison ... were the truth). This my strongest comment on the paper: Since the requirements on XCO2 are so stringent, it matters for the comparisons in this paper how exactly 1) the HIPPO profiles are extended, 2) the averaging kernel is applied, and 3) the null-space is attributed. I would recommend to incorporate a small section/paragraph explaining the mathematical details.

Questions that come to mind: Is the model information just attached to the HIPPO profile? If a jump would appear in such a profile, how is that treated? Is the smoothed (extended) HIPPO profile compared to the GOSAT profile without null-space contribution, or is there also a null-space contribution to the smoothed HIPPO profile? If the latter, which reference is used? The same as in the GOSAT retrievals, or the model?

p11, l22-24: Even after ... for MACC. Please refer to Figs 5 and 6

p11, l22: Even after normalization It is clear how the HIPPO data is corrected, but how is the other data corrected? With the HIPPO value, or with the average value of the particular model?

p13, l23: lower left quadrant Maybe the authors would like to note that these points are also outliers in the CT comparison. Not as strong as in the case of MACC, but still in

the same quadrant, which may be an indication that the transport errors in both models are roughly equal and/or the GFED data is somewhat off.

p24, Fig 3: There are some strong excursions in the HIPPO profiles close to the surface; any explanation for these? For HIPPO-1, 3, and 4 (and possibly 5), the differences between HIPPO and MACC resp. CT differs significantly for > 70N. Any explanation for this behaviour? Please, reposition the legend box; CT-HIPPO 5 is barely visible.

p26-p28, Fig 5-7: Mention the shift for both axes

p31,p32, Fig 10,11: Mention the shift for both axes

Technical corrections:

p4, l5: Greenhouse Gas Observing –> Greenhouse Gases Observing

p4, l5: haven –> have

p4, l11: sensing measurement –> sensing measurements

p5, l5-8: This sentence does not have a verb. Suggestion: The HIAPER Polo-to-Pole Observations (HIPPO) project consists of a sequence of ...

p6, l23: LSCE. To be on the safe side I would explicitedly write out this acronym

p9, l21-22: consistent between model, –> consistent between the two models,

p10, l6: usually 162 253 –> usually

There are several places where ppb is used in stead of ppm: p11, l24 p11, l25 p21, Table 1 (2 instances)

p16, l11: that –> than

---

## Author Comment (AC1) · 10 May 2016

We thank Anonymous Referee #1 for a positive and thorough review. In the following, we will respond to the Reviewers comments step by step.

Minor issues: Page 10, line 24: I think "an excellent agreement" is a bit too strong taking into account that there are differences up to 4 ppm (same page, line 20). I recommend to change this to "good agreement" or so.

→ Changed to "good". We also removed the qualitative comparison statement between MACC and CT2013B as it wasn't really too substantiated

Similar on page 11, line 21, with the statement "compare extremely well" already in the first sentence before any comparison results are shown and discussed. I also do not think that agreement within 1 ppm and outliers up to 3 ppm is best characterized by "extremely well" (NOTE: in the text on page 11 the unit ppb is given two times (lines 24 and 25) but I guess this should be ppm!).

→ Removed "extremely" and changed ppb to ppm (old methane habit).

Page 10, line 25: "In some cases, MACC seems to compare somewhat better, . . .". A MACC colleague is co-author but no CarbonTracker colleague. I wonder if NOAA would agree with this statement. I also wonder if NOAA needs to be acknowledged for their data.

→ We removed this statement altogether and added an acknowledgement to that effect, esp. as Andy Jacobson was involved in our discussions but not listed as co-author.

Page 11, line 6: "SCIAMACHY data over the oceans is not yet matured as is has no dedicated Glint mode." Sounds a bit strange (even if "is" typo corrected). I recommend to replace this with "SCIAMACHY data have not been used as it has no dedicated glint mode and the SCIAMACHY products (e.g., Reuter et al., 2011) are limited to retrievals over land". Reuter et al., 2011: "Retrieval of atmospheric CO2 with enhanced accuracy and precision from SCIAMACHY: Validation with FTS measurements and comparison with model results", J. Geophys. Res.

→ done

Page 16, line 8: ". . . indicates that GOSAT compares slightly better overall." Compared to what?

→ removed that sentence and added "comparable to those with models" to the prvious sentence.

Page 21, Tab. 1: Why is the GOSAT sigma only 0.45 ppm (as far as I know the GOSAT XCO2 single measurement precision is about 2 ppm; or have data been averaged?)? Please check and add additional explanation if necessary.

→ Yes, multiple GOSAT soundings are used per HIPPO profile and averaged (as stated before, "For the GOSAT comparison, we require more than 5 co-located GOSAT measurement per HIPPO profile.". We changed that sentence to

"For the GOSAT comparison, we require at least 5 co-located GOSAT measurement per HIPPO profile, all of which are subsequently averaged before comparison against HIPPO". It was also stated before that "For each match, the standard error in the GOSAT XCO2 average is computed using the standard deviation of all corresponding GOSAT colocations divided by the square root of the number of colocations."

Fig. 3: Bottom, middle: Profiles only partially visible as overplotted by legend. Please improve.

→ done

Fig. 5, left: Possibly data points only partially visible as overplotted by legend. Please improve.
→ done

Fig. 6, left: Data points only partially visible as overplotted by legend.
→ done

Please improve. Fig. 7, right: Symbols for models very difficult to see in printout.
→ We would ask the editorial office to check into that issue.

No reference to Figs. 5 and 6 in text (should be somewhere in Sect. 4).
→ Added "In terms of XCO$_2$, both atmospheric models used here compare well against HIPPO, as can be seen in Figures 5 and 6. (at beginning of Sec. 4). Thanks for noticing this!

No reference to Fig. 11 in text (should be somewhere in Sect. 5.2).
→ Added, thanls

Typos: Page 10, line 6: "are usually 162253" ?
→ Typical LaTeX typing error (accidentally copying something without noticing it), apologies.
Caption Tab. 1: "of different compared to" -> "of the difference compared to"
→ Done, thanks

---

## Author Comment (AC2) · 10 May 2016

We thank Anonymous Referee #2 for a positive and thoughtful review. In the following, we will respond to the Reviewers comments step by step.

Minor revisions: p5, l8-10: This enables ... denoted XCO2 I think this statement would be more clear to the reader when a line is added to indicate that an extension is needed above 14 km. Then you can indeed state that this extension is of limited consequence since most of the variability in XCO2 stems from the troposphere which is covered by the HIPPO profiles.

- ➔ Good point. We changed to: "This enables a comparison of individual sub-columns of air but also of column-averaged mixing ratios of CO\$\_2\$, denoted XCO\$\_2\$, if the profile can be reliably extended above 14\,km. As the troposphere dominates the variability in XCO\$\_2\$, errors induced by extending profiles are expected to be small."
- p7, l11-12: As most ... analysis. Add a line why SCIAMACHY does not provide data over oceans
  Added "...because it lacks a dedicated Glint measurement mode" and explained it better later as well, as requested by Rev. #1.
- p7, l16: short-wave −> short-wave infrared→ done
- p8, l11: How can averaging lead to the reduction of systematic errors?
  - → Removed systematic here.

p9, l9-11: Validation ... (Olsen and Licata, 2014). If Olsen and Licata already have compared IR/MW L2 and IR-Only L2 against HIPPO, then I would expect a sentence explaining how the current study differs and/or extends wrt. the cited paper.

→ We rephrased and extended that sentence to reflect the main differences (using models to fill up the profile).: Olsen and Licata (2014) compare the IR/MW based and IR-Only based CO2 retrievals over the globe for 2010-2011 and for collocations with the deep-dip HIPPO-2, HIPPO-3, HIPPO-4 and HIPPO-5 profiles. Their global analysis reveals that the zonal monthly average difference rarely exceeds 0.5 ppm save at the high northern latitudes in January and October where fluctuations resulting from small number statistics dominate. Their analysis against HIPPO employs only the deep-dip measured profiles, i.e. those in which the aircraft reached the 190 hPa pressure level, to ensure good in situ measurement coverage of the AIRS sensitivity profile and to minimize the error introduced by their simple approximation of extending the aircraft profile into the stratosphere by replicating the highest altitude measurement. During the HIPPO-2 and HIPPO-3 campaigns, the AMSU channel 5 noise figure was acceptable, whereas during the HIPPO-4 and HIPPO-5 campaigns it progressively degraded at a rapid rate. For all campaigns, the two sets of collocations, averaging AIRS retrievals within ±24 hours and 500 km of the aircraft profile, exhibit the same bias and RMS to within 1 ppm for llat  $\leq$ 60°. The current study extends the in situ measurements to higher altitude by the means of CarbonTracker and MACC model output, thereby allowing use of all HIPPO profiles rather than only the deep-dip profiles. Our results are statistically consistent with the

latitude-dependent biases reported by Olsen and Licata (2014) and give a more detailed view of the scatter as a function of latitude.

p9, l14-18: For the differences ... should dominate If you first extend the HIPPO profile with model data, then integrate, and finally subtract the integrated model data, does the part above flight altitude not exactly cancel? HIPPO (< 14 km) + model (> 14 km) - model (0-TOA) = HIPPO (< 14 km) + model (> 14 km) - model (> 14 km) - model (< 14 km) - model (< 14 km) - model (> 14 km) - model (< 14 km) - model (> 14 km) - model (< 14 km) - model (> 14 km) - mod

→ About 20% of the total column is located above 14km and not all HIPPO profiles extended that far. If we use part of the model, these values indeed cancel and yield exactly 0 difference, making the agreement somewhat better. With the 80/20 weighting, it is similar to saying that delta-XCO2 is 0.8\*(Model-HIPPO)+0.2\*(model-model=0), thus potentially always dampening the differences. Or, if the profile extended only to 10km, dampening it even further.

p10, l14-23: Figure 4 ... potential biases. HIPPO 3 is nicely explained in this paragraph, but HIPPO 5 is depicted in the Figure but not mentioned. Any comment that the authors can make on the MACC and CT differences/similarities?

→ Added "In HIPPO 5, at the end of the growing season, the situation is reversed as the profile slopes change sign after the large CO\$\_2\$ uptake during summer." And "For HIPPO 5, the deviations for CT2013B are somewhat smaller but it can be seen that most models suffer from these potential biases if large vertical gradients exist."

p11, I2-10: Here, we look ... in the future. This alinea is mostly about measurements and campaigns that are not treated in the paper. I understand why the authors like to mention this, but maybe the conclusion, which includes a future outlook mentioning OCO-2, is the better spot for this.

→ This is indeed better, we moved this to the Conclusions.

p11, l11-19: For the comparison ... were the truth). This my strongest comment on the paper: Since the requirements on XCO2 are so stringent, it matters for the comparisons in this paper how exactly 1) the HIPPO profiles are extended, 2) the averaging kernel is applied, and 3) the null-space is attributed. I would recommend to incorporate a small section/paragraph explaining the mathematical details. Questions that come to mind: Is the model information just attached to the HIPPO pro- file? If a jump would appear in such a profile, how is that treated? Is the smoothed (extended) HIPPO profile compared to the GOSAT profile without null-space contribution, or is there also a null-space contribution to the smoothed HIPPO profile? If the latter, which reference is used? The same as in the GOSAT retrievals, or the model?

→ This is a good point even though we prefer to keep this short in the paper. Re 1). The HIPPO profiles are extended with the model data before applying the averaging kernel correction. 2). The AK corrected HIPPO values are computed as xa+A(xt-xa) with the a priori profile xa and the "true" profile xt (HIPPO + model). For GOSAT, the column averaging kernel was used, for TES and AIRS the averaging kernel for the respective retrieval layer.

We have not tested the impact of a jump in a profile; in the manuscript, a simple profile extension was performed without testing smoothness. In most cases, the impact should be relatively small. The null space contribution in GOSAT comparisons should be small as the column averaging kernels are relatively large throughout the entire column. In general, HIPPO data has always been filled in with model data, not satellite priors. We added

For GOSAT: "For the HIPPO comparison against GOSAT data, we take the instrument sensitivity into account by applying the averaging kernel to the difference of the true profile (using the model-extended HIPPO dataset as truth) and the respective a priori profile. We perform this correction using both model extensions independently and then use the average of the two. "

For TES: "For the comparison with TES, we use the 510\,hPa retrieval layer and apply averaging kernel corrections using model-extended HIPPO data as {\em truth}, using both models indepdently and averaging results after averaging kernel correction." For AIRS: "For the comparison with AIRS (Fig\,. \ref{fig:HIPPO\_AIRS}), the sensitivity maximum varies around 300\,hP and we apply the averaging kernels similarly to TES." We hope this will clarify the issue.

p11, l22-24: Even after ... for MACC. Please refer to Figs 5 and 6

**➔ done**

p11, l22: Even after normalization It is clear how the HIPPO data is corrected, but how is the other data corrected? With the HIPPO value, or with the average value of the particular model?

→ With the HIPPO value. We added a sentence "For each campaign, we also normalize all data with the respective campaign average of the HIPPO dataset."

p13, l23: lower left quadrant Maybe the authors would like to note that these points are also outliers in the CT comparison. Not as strong as in the case of MACC, but still in C3 the same quadrant, which may be an indication that the transport errors in both models are roughly equal and/or the GFED data is somewhat off.

→ We mentioned that "both models" show that feature.

p24, Fig 3: There are some strong excursions in the HIPPO profiles close to the surface; any explanation for these?

→ These might be caused by dips close to the surface with HIPPO, potentially coming from the land data. It should not really affect XCO2 a lot as it only affects a small subcolumn.

HIPPO-1, 3, and 4 (and possibly 5), the differences between HIPPO and MACC resp. CT differs significantly for > 70N. Any explanation for this behaviour?

→ We agree, there seem to be substantial differences but we don't have any explanation yet for this and would not like to speculate too much.

Please, reposition the legend box; CT-HIPPO 5 is barely visible.

➔ done

p26-p28, Fig 5-7: Mention the shift for both axes

- → we now state "Scatterplot of normalized (with campaign average) CO2..."
- p31,p32, Fig 10,11: Mention the shift for both axes

→ see above

- p4, I5: Greenhouse Gas Observing -> Greenhouse Gases Observing
  - ➔ don
- p4, I5: haven -> have
  - ➔ fixed, thanks.
- p4, l11: sensing measurement → sensing measurements
   → done

p5, I5-8: This sentence does not have a verb. Suggestion: The HIAPER Polo-to-Pole Observations (HIPPO) project consists of a sequence of ...

- → replaced "sampling" with "sampled"
- p6, I23: LSCE. To be on the safe side I would explicitly write out this acronym → Done

p9, l21-22: consistent between model, -> consistent between the two models,

➔ done

p10, l6: usually 162 253 → usually → done

There are several places where ppb is used in stead of ppm: p11, l24 p11, l25 p21, Table 1 (2 instances)

➔ done

p16, l11: that -> than

➔ done, thanks